# The Longitudinal Analysis on the Anti-SARS-CoV-2 Antibodies among Healthcare Workers in Poland—Before and after BNT126b2 mRNA COVID-19 Vaccination

**DOI:** 10.3390/vaccines10101576

**Published:** 2022-09-20

**Authors:** Dagny Lorent, Rafał Nowak, Dawid Luwański, Magdalena Pisarska-Krawczyk, Magdalena Figlerowicz, Paweł Zmora

**Affiliations:** 1Institute of Bioorganic Chemistry Polish Academy of Sciences, 61-704 Poznan, Poland; 2Gynecology and Obstetrics Ward, District Hospital in Wrzesnia, 62-300 Wrzesnia, Poland; 3Department of Nursery, The President Stanislaw Wojciechowski Calisia University, 62-800 Kalisz, Poland; 4Department of Infectious Diseases and Child Neurology, Poznan University of Medical Sciences, 61-701 Poznan, Poland

**Keywords:** SARS-CoV-2, antibodies, healthcare workers, seroprevalence, vaccine, Poland

## Abstract

One of the groups most vulnerable to severe acute respiratory syndrome coronavirus 2 (SARS-CoV-2) infection is healthcare workers (HCWs) who have direct contact with suspected and confirmed coronavirus diseases 2019 (COVID-19) patients. Therefore, this study aimed to (i) conduct a longitudinal analysis of the seroprevalence of SARS-CoV-2 infection among HCWs working in two healthcare units (HCUs) in Poland and (ii) identify anti-SARS-CoV-2 IgG antibody (Ab) response factors following infection and anti-COVID-19 vaccination. The overall seroprevalence increased from 0% at baseline in September 2020 to 37.8% in December 2020. It reached 100% in February 2021 after BNT126b2 (Pfizer New York, NY, USA/BioNTech Mainz, Germany) full vaccination and declined to 94.3% in September 2021. We observed significant differences in seroprevalence between the tested high- and low-risk infection HCUs, with the highest seropositivity among the midwives and nurses at the Gynecology and Obstetrics Ward, who usually have contact with non-infectious patients and may not have the proper training, practice and personal protective equipment to deal with pandemic infections, such as SARS-CoV-2. We also found that anti-SARS-CoV-2 Ab levels after coronavirus infection were correlated with disease outcomes. The lowest Ab levels were found among HCWs with asymptomatic coronavirus infections, and the highest were found among HCWs with severe COVID-19. Similarly, antibody response after vaccination depended on previous SARS-CoV-2 infection and its course: the highest anti-SARS-CoV-2 Ab levels were found in vaccinated HCWs after severe COVID-19. Finally, we observed an approximately 90–95% decrease in anti-SARS-CoV-2 Ab levels within seven months after vaccination. Our findings show that HCWs have the highest risk of SARS-CoV-2 infection, and due to antibody depletion, extra protective measures should be undertaken. In addition, in the context of the emergence of new pathogens with pandemic potential, our results highlight the necessity for better infectious disease training and regular updates for the low infection risk HCUs, where the HCWs have only occasional contact with infectious patients.

## 1. Introduction

Since the first reported case of severe acute respiratory syndrome coronavirus 2 (SARS-CoV-2) in December 2019 in Wuhan, China, the WHO has reported almost 597 million confirmed cases of novel coronavirus infections and 6.46 million deaths related to coronavirus diseases 2019 (COVID-19) [1]. One of the groups most vulnerable to SARS-CoV-2 infections is healthcare workers (HCWs) who have direct contact with suspected and confirmed COVID-19 patients [2,3,4]. The best way to prevent a novel coronavirus infection is vaccination [5,6,7,8,9,10]. Safe and efficient COVID-19 vaccines were developed and approved by the U.S. Food and Drug Administration (Silver Spring, MD, USA) and the European Medicines Agency (Amsterdam, The Netherlands) in 2020 [11,12]. Since the healthcare system is the most crucial element in the fight against COVID-19 and depends completely on HCWs, the WHO recommended vaccinating HCWs during the first phase of vaccination campaigns [13].

The Polish vaccination strategy included four phases [14]. So called phase 0 included COVID-19 vaccination with two doses of BNT126b2 mRNA vaccine (Pfizer/BioNTech) restricted to the HCWs, medical students and social care workers. The first vaccines were administered in the last days of December 2020, and as of March 2021, most HCWs had been fully vaccinated. In phase 1, which started in mid-January 2021, COVID-19 vaccines were offered to elderly over 60 y.o., long-term care facilities residents and public service workers, i.e., teachers. During the next phase, individuals with comorbidities, as well as other essential workers might be vaccinated. Finally, in phase 3 the COVID-19 vaccine might be administered to each person older than 18 y.o. In phases 1–3, the following COVID-19 vaccines were available: BNT126b2 mRNA vaccine (Pfizer/BioNTech), mRNA-1273 (Moderna, Cambridge, MA, USA), AZD1222 (AstraZeneca, Cambridge, UK/Oxford University, Oxford, UK) and JNJ-78436735 (Janssen Pharmaceutical Companies, Beerse, Belgium). Depending on the current epidemiological situation and vaccine availability, the vaccination rollout was updated, with specific COVID-19 vaccines targeting particular groups, i.e., the AZD1222 with a 3-month time period between the doses was offered to the teachers. Up to date, approximately 57.8% of the Polish society was fully vaccinated [15], and the HCWs are one of the groups with the highest vaccination rate. It is estimated that approximately 93% of physicians, 90% of dentists, 92% of medical laboratory assistants, 90% of midwives and 89% of nurses administered at least two doses of the COVID-19 vaccine [16].

The BNT162b2 mRNA COVID-19 vaccine, developed by Pfizer and BioNTech, consists of nucleoside-modified mRNA encoding full-length SARS-CoV-2 spike (S) glycoprotein formulated in lipid nanoparticle composition. This vaccine showed high efficacy in preventing COVID-19 two and six months after the second vaccine dose, i.e., 95% [17] and 91.3% [18], respectively. However, it should be highlighted that vaccine protection is not endless and a reduction in effectiveness against COVID-19 and waning immunity with time was observed [19]. The seroprevalence of SARS-CoV-2 infection, as well as antibody response after COVID-19 vaccination among HCWs in Poland [20,21,22], as well as other countries [3,23,24] were analyzed by many research groups. The immune response to natural SARS-CoV-2 infection, as well as COVID-19 vaccination, is not fully understood.

Here, we present the antibody responses after natural SARS-CoV-2 infection and COVID-19 vaccination among HCWs from different healthcare units with different SARS-CoV-2 infection risks at four time points through one year.

This study aimed to (i) conduct a longitudinal analysis of the prevalence of anti-SARS-CoV-2 antibodies among HCWs in Poland and (ii) identify anti-SARS-CoV-2 IgG antibody response factors following coronavirus infection and anti-COVID-19 vaccination.

## 2. Materials and Methods

### 2.1. Study Participants

We invited HCWs from several COVID-19 and non-COVID-19 hospital healthcare units (HCUs) in the Greater Poland region, Poland. Based on healthcare unit specificity, such as structure, size, and numbers of employees and patients, we included HCWs from the Department of Infectious Diseases and Child Neurology (DIDaCN), K. Jonscher’s Clinical Hospital, Poznan University of Medical Sciences, Poznan, and the Gynecology and Obstetrics Ward (GaOW), District Hospital, Wrzesnia, in the study. The HCWs were invited to participate in the project on a voluntary basis.

### 2.2. Study Design

All participants were asked to answer an online epidemiological survey to collect data on sex, age, profession, and previous SARS-CoV-2 RT-PCR test results. Additionally, HCWs with confirmed COVID-19 were questioned about the severity of their symptoms. Individual flu-like symptoms, e.g., fever, cough, runny nose, fatigue, and muscle and joint pain, were rated as mild to moderate, while participants who required hospitalization were classified as having a severe course of infection. HCWs who had not shown any flu-like symptoms in the last nine months before the analysis but had received positive results for an anti-SARS-CoV-2 ELISA were defined as asymptomatic.

Blood samples were collected from 90 individuals at either the DIDaCN or GOaW in September 2020 after the first wave of the COVID-19 pandemic; in December 2020 after the second wave of the COVID-19 pandemic; in February 2021, approximately two weeks after the second dose of the BNT162b2 (Pfizer/BioNTech) vaccine; in September 2021 after the third wave of the COVID-19 pandemic and approximately seven months after vaccination (Appendix A). At the last time point, 15 out of 90 enrolled individuals did not donate their blood since they did not show up and did not answer our contact attempts.

### 2.3. Laboratory Analysis

Before the COVID-19 vaccination, we determined the prevalence of SARS-CoV-2 infection using semi-quantitative anti-SARS-CoV-2 S IgG ELISA (EuroImmun GmbH, Lübeck, Germany). In addition, positive ELISA results were confirmed by quantitative anti-SARS-CoV-2 IgG immunoblots (Polycheck; Biocheck GmbH, Münster, Germany), which detected the anti-SARS-CoV-2 spike (S) protein and nucleocapsid protein (NCP) antibodies. After COVID-19 vaccination, a quantitative analysis of anti-SARS-CoV-2 S antibody levels was performed using anti-SARS-CoV-2 QuantiVac ELISA IgG (EuroImmun GmbH) according to the manufacturer’s instructions. Moreover, to analyze the SARS-CoV-2 seroprevalence after vaccination we used the anti-SARS-CoV-2 NCP IgG ELISA (EuroImmun GmbH), which detects anti-SARS-CoV-2 NCP antibodies generated only after natural infection and not produced as a consequence of BNT126b2 mRNA COVID-19 vaccination. The semiquantitative anti-SARS-CoV-2 NCP ELISA results were further confirmed by the quantitative anti-SARS-CoV-2 IgG immunoblots (Polycheck, Biocheck GmbH).

### 2.4. Statistical Analysis

The categorical variables were presented as counts and percentages, and the seroprevalence estimates were presented with the 95% CI calculated using the hybrid Wilson/Brown method. The differences between the two groups were analyzed by the Mann–Whitney test. For the analysis of the seroprevalence at different time points, a repeated measures ANOVA with the Tukey post-hock test was performed with HCUs as a between-subject factor and time of blood donation as a within-subject factor. Further analyses were conducted with two-way ANOVA with the Tukey post-hock test with the following within-subject factors: sex (female, male); age (<30, 31–40, 41–50, 51–60, >60); occupation (physician, nurse/midwife, other); time after infection (3, 4, 5, etc.); COVID-19 course (asymptomatic, mild to moderate, severe), ELISA result before vaccination (negative, positive). The interaction between analyzed factors was not included. Data were accepted as statistically different if *p* < 0.05. All statistical analyses were performed using GraphPad Prism 9 software.

### 2.5. Ethics Approval

The study was approved by the Bioethics Committee at the Poznan University of Medical Sciences, Poznan, Poland (Resolution No. 470/20 from 17 June 2019). In addition, written informed consent was obtained from each study participant before blood collection.

## 3. Results

### 3.1. Characteristics of the Study Participants

The study group consisted of 90 HCWs from two HCUs in the Greater Poland region, i.e., 50 participants from the DIDaCN and 40 participants from the GaOW (Table 1). Most of the enrolled HCWs were female: 80% (n = 40) from the DIDaCN and 87.5% (n = 35) from the GaOW. Participants from the DIDaCN were aged 36.5 ± 10.8 years; 50% were nurses (n = 25), 32% were physicians (n = 16), and 18% were other employees (n = 9). Participants from the GaOW were aged 53 ± 10.6 years; 72.5% of them were midwives (n = 29), 15% were physicians (n = 6), and 12.5% were other employees (n = 5) (Table 1). At the time of enrolment, none of the HCWs had a history of COVID-19.

Although the HCUs were located in two different cities in the Greater Poland region, i.e., Poznan and Wrzesnia, they were of a similar size with a similar number of employees (Table 1). The DIDaCN had 20 hospital beds and had admitted approximately 273 patients/month in 2020. The GaOW had 28 hospital beds and three delivery rooms and had admitted approximately 150 women/month in 2020. However, the HCUs differed significantly in the number of patients admitted with suspected/confirmed COVID-19 (Table 1). The GaOW only admitted three SARS-CoV-2 positive patients throughout 2020, while the DIDaCN admitted 292 confirmed and 2443 suspected COVID-19 patients in 2020. Due to suspected and confirmed contact with COVID-19 patients, we classified the DIDaCN as a high-risk SARS-CoV-2 infection work environment in contrast to the GaOW, which we defined as a low-risk infection environment (Table 1).

### 3.2. Prevalence of SARS-CoV-2 Infection among HCWs

At the time of enrolment (September 2020), none of the participants had a history of COVID-19, and this was confirmed by the ELISA results. We did not detect any anti-SARS-CoV-2 antibodies in any of the participants. However, over the following three months (October–December 2020), as a second wave of the pandemic affected Poland, total seroprevalence among the participants increased to 37.8% (95% CI 28.46–48.10). However, there were some differences between the HCUs: there were twice as many participants with anti-SARS-CoV-2 antibodies at the GaOW compared to the DIDaCN (Table 2). Based on the self-assessments of the ELISA-positive participants, we categorized six as having asymptomatic SARS-CoV-2 infection; 24 as having mild to moderate COVID-19 and three, who required hospitalization, as having severe COVID-19.

At the beginning of 2021, COVID-19 vaccines started to become available, and according to WHO recommendations, HCWs in Poland were vaccinated during the phase 0 of the vaccination campaign. After two doses of the BNT162b2 mRNA vaccine (Pfizer-BioNTech), anti-SARS-CoV-2 S antibodies were detected in all the study participants (Table 2). Due to the presence of antibodies against SARS-CoV-2 S after vaccination, we decided to further analyze the prevalence of SARS-CoV-2 infection based on the anti-SARS-CoV-2 NCP antibodies.

In February 2021, we observed a significant decrease in the total prevalence of SARS-CoV-2 infection (Table 2), and anti-SARS-CoV-2 NCP antibodies were found among 68.7% of previously ELISA-positive individuals. It should be highlighted that, simultaneously, none of the ELISA-negative individuals seroconverted and received positive results at this time. In addition, a significantly higher seroprevalence was observed among the GaOW participants compared to the DIDaCN participants (Table 2).

Within the next seven months (as of September 2021), we observed further depletion of anti-SARS-CoV-2 NCP antibodies. In addition, the vaccination antibodies were completely depleted in some individuals, from both the DIDaCN and the GaOW. Similar to the previous observation, we only detected antibodies in previously ELISA-positive participants. None of the previously analyzed ELISA-negative participants were infected with SARS-CoV-2.

### 3.3. The SARS-CoV-2 Infection Risk Factors among HCWs

Based on the obtained seroprevalence results and the study participant’s questionnaires we analyzed the impact of some general factors, such as sex, age, and profession on the presence of anti-SARS-CoV-2 antibodies. We did not observe any significant differences in the seroprevalence between females and males, as well as nurses/midwives and physicians (Table 3). Additionally, the seroprevalence in most age groups is similar, with one exception. Namely, the highest SARS-CoV-2 seroprevalence (51.7%, 95% CI 14.55–51.90) was found among study participants between 50 and 60 years old (Table 3).

In addition, we analyzed the SARS-CoV-2 infection risk factors in high- and low-infection risk environments. At the DIDaCN, we found the highest seropositivity for participants working as physicians (31.3%, 95% CI 14.17–55.60), followed by nurses (24%, 95% CI 11.50–43.43) and other employees (22.2%, 95% CI 3.95–54.74). Among the GaOW’s medical personnel, the occupation group with the highest seropositivity included nurses/midwives (62.1%, 95% CI 44.00–77.31%), followed by physicians (50%, 95% CI 18.76–81.24%). Note, that nurses and midwives from the GaOW had 2.5 times higher seropositivity than those from the DIDaCN.

Since no new SARS-CoV-2 infections among HCWs were not observed after the second pandemic wave and COVID-19 vaccination, we did not analyze the infection risk factors at later time points.

### 3.4. Anti-SARS-CoV-2 Antibodies Level after the Infection

The anti-SARS-CoV-2 IgG antibody levels varied widely across the participants, ranging from 0.73 to 139 kU/L for anti-S and from 0.73 to 119 kU/L for anti-NCP. In addition, we did not observe any correlation between anti-S and anti-NCP antibody levels (data not shown). For all the participants, there were no significant differences in the anti-SARS-CoV-2 IgG S or NCP antibody levels by sex, age, or time after infection (Figure 1a–c). However, we found that the anti-SARS-CoV-2 IgG levels after coronavirus infection were correlated with the disease course. For both anti-SARS-CoV-2 IgG S and IgG NCP antibodies, participants with prior symptomatic SARS-CoV-2 infection had significantly higher antibody levels than asymptomatic participants (Figure 1d). Moreover, we observed that the anti-NCP antibody levels in participants with mild to moderate COVID-19 symptoms could be divided into three subpopulations. However, as the course of the SARS-CoV-2 infection was based on the participants’ personal assessments, there was a possibility that a more precise trend would have arisen if a more objective evaluation of COVID-19 symptom severity had been made.

### 3.5. Anti-SARS-CoV-2 Antibodies Level after Vaccination

Based on the WHO’s recommendation and according to the Polish vaccination campaign strategy, HCWs were vaccinated during the phase 0 of the campaign. After two doses of the vaccine, 100% of the study participants developed anti-SARS-CoV-2 IgG S antibodies, ranging from 287 to 19,363 BAU/mL. We did not observe any effect of sex or age on the anti-SARS-CoV-2 S antibodies (Figure 2a,b). However, we found that the anti-SARS-CoV-2 IgG S antibody level after vaccination was significantly higher among participants previously infected with SARS-CoV-2 compared to infection-naive participants (Figure 2c). In addition, the antibody response was correlated with COVID-19 severity, i.e., participants with mild to severe symptoms had significantly higher anti-SARS-CoV-2 IgG S antibody levels than asymptomatic participants, and the highest antibody levels were found among individuals who had severe COVID-19 (Figure 2d).

Due to the previously observed decrease in the prevalence of SARS-CoV-2 infection among the participants, we decided to analyze for vaccination antibodies approximately seven months after the second dose of the BNT162b2 vaccine. Simultaneously, we analyzed for changes in the anti-SARS-CoV-2 NCP antibody level by quantitative immunoblot. We observed an approximately 90–95% decrease in the anti-SARS-CoV-2 S antibodies developed after vaccination (Figure 3). Moreover, we did not find any effect of sex, age and previous SARS-CoV-2 infection on the antibody depletion rate (Appendix A). In addition, we did not detect any vaccination antibodies among the four participants. Similarly, the anti-SARS-CoV-2 NCP antibodies were completely depleted from the 19 previously COVID-19 positive participants, and among the rest of the previously SARS-CoV-2 infected participants, the antibody levels were approximately 85% lower compared to February 2021. Interestingly, for unknown reasons, we observed an increase in vaccination and post-infection antibodies among two and four individuals, respectively.

## 4. Discussion

Our previous study demonstrated that the highest SARS-CoV-2 infection risk in Poland was related to age, compliance with epidemiological recommendations, e.g., face mask, hand disinfection and social distancing, traveling abroad, and occupations related to constant contact with people, such as healthcare workers [4]. In light of this and to guarantee constant medical services, longitudinal analyses of COVID-19 cases among HCWs and estimations of SARS-CoV-2 infection in naïve HCWs should be performed on a regular basis. In the current study, we analyzed SARS-CoV-2 seroprevalence among HCWs at different times related to pandemic waves. However, we did not identify any HCWs with anti-SARS-CoV-2 antibodies after the first wave, i.e., May–September 2020. Very low seroprevalence among medical personnel at the same time was also noted in different Polish cities, i.e., Warsaw 0.85% [20], Opole 1.1% [21], and Lublin 2.4% [22], and low numbers of COVID-19 cases among HCWs after the first months of the pandemic were found in different countries, i.e., Germany 1.6% [24], Greece 1.3% [25], and the USA 1.1% [26,27]. This can be explained by low seroprevalence in the general population after the first wave of the pandemic, i.e., the Greater Poland region 0.97% [4], Greece 0.36% [28], and the USA 1% [29,30]. The low incidence of SARS-CoV-2 at the beginning of the pandemic implies that the population had little herd immunity heading into the second wave of the pandemic and may explain the dramatic increase in the number of COVID-19 cases, which also occurred among HCWs in later months. After the second wave of the pandemic, i.e., November 2020–January 2021, which led to 1.4 million confirmed COVID-19 cases and 33.4 thousand COVID-19-related deaths [15], we observed a significant increase in seroprevalence among enrolled HCWs. A similar observation was made among HCWs from different Polish cities, e.g., Lublin [22]; however, this seroprevalence was two times lower. These discrepancies can be explained by the differences in the number of confirmed COVID-19 cases at the end of the second wave of the pandemic in the Greater Poland and Lublin regions, 132,423 and 66,027, respectively [15]. Additionally, the prevalence of SARS-CoV-2 in the general population increased to between 11.2% and 22.9% [31], although this was to a lower extent compared to the analyzed HCWs, which confirmed our previous finding that HCWs have the highest risk for SARS-CoV-2 [4]. At later time points, i.e., after COVID-19 vaccinations, which were first administered to HCWs, we did not observe any new cases of SARS-CoV-2 infections among HCWs, while overall seropositivity in the Polish population grew dramatically. This decrease in the HCW’s seroprevalence may be explained by antibody depletion and the effectiveness of the vaccine against infection, which was also demonstrated by others [32,33,34].

Differences in SARS-CoV-2 seroprevalence among HCWs have been demonstrated by many research groups. For example, a meta-analysis by Kayi et al. [35] indicated that the most common risk factors associated with higher seroprevalence rates were ethnicity, male gender, and a high number of household contacts, while working in a high-risk infection environment did not affect seropositivity. Galanis et al. [3] meta-analysis of COVID-19 risk factors defined as gender and household contacts additionally showed that working in a COVID-19 unit or as a front-line HCW was also associated with seropositivity. The clinical work environment, i.e., with high-risk exposure, was also highlighted as an important factor for seroprevalence in a cross-sectional study by Ebinger et al. [36] and Piccoli et al. [23]. In our study, we found that HCWs from low-risk infection environments had higher rates of seropositivity, which may be explained by the specificity of the work at the GaOW, including not enough time to perform COVID-19 diagnostics when pregnant women have started childbirth, the lower availability of and experience using personal protective equipment, and the uncontrolled behavior of women during labor contractions. Our observations confirmed findings demonstrated by Alishaq et al. [37] and Bampoe et al. [38], i.e., unusually high rates of SARS-CoV-2 infections among the maternity ward personnel, including breakthrough infections following vaccination. It should be highlighted that in the gynecology and obstetrics wards HCWs are used to working with non-infectious patients, and do not have proper knowledge, practice and experience to handle patients with COVID-19 for example [39,40]. Therefore, some additional infectious diseases training, regular updates, as well as full availability of the personal protective equipment for the HCWs from the low infection risk HCUs are extremely important and should be included in the healthcare system maintenance. In addition, the higher seroprevalence at the GaOW can also be explained by the higher number of COVID-19 cases in Wrzesnia per 100,000 citizens compared to Poznan, that is, 1179.29 and 447.28, respectively (data from the end of the second pandemic wave in January 2021) [15]. These results are similar to those reported by Russel et al. [41], who observed that the local increase in COVID-19 cases may contribute to the higher seroprevalence among healthcare workers.

As a result of pathogen infection, the human body activates an immune response and, among others, starts producing pathogen-specific antibodies. The level of antibody response depends on the type of pathogen and its ability to inhibit and delay the human immune response, the age and gender of the infected patient, genetic predisposition, and overall health status, i.e., existing comorbidities [42], as well as lifestyle, diet, and physical activity levels [43]. However, with vaccination, humans can produce pathogen-specific antibodies that protect them from diseases they have not been previously infected with, which can sometimes lead to severe outcomes, patient health complications, and even death. The antibody response after vaccination depends on (i) vaccine factors, such as type, i.e., attenuated, inactivated, genetic, etc., dose, and adjuvant presence; (ii) administration factors, such as vaccination schedule, time of vaccination, and the number of doses and the periods between them; (iii) host factors, i.e., age, sex, genetics, and existing comorbidities; (iv) behavioral factors, i.e., smoking, alcohol consumption, exercise, and sleep and (iv) nutritional factors, i.e., body mass index, micro- and macronutrients, and potential enteropathy [44]. In our study, we mostly focused on the host and environmental factors that affect post-infection and post-vaccination antibody responses. In both cases, we found very heterogeneous levels of antibodies and did not observe any association between the antibody level and the sex or age of the participants. Similar findings have been published by many scientists [45,46,47,48], while others have reported significant differences in anti-SARS-CoV-2 antibodies between the sexes and people of different ages. These very heterogeneous levels were also found in the relatively low number of COVID-19 convalescent study participants who donated their blood at different times after infection. Similar results were published by Moncunill et al. (2022) [49]. The number of participants and the overrepresentation of either men or women in the study groups may have contributed to these discrepancies.

According to the WHO, vaccination is the best and safest way to protect against infectious diseases. Vaccines last longer and are more effective in producing antibodies than those acquired following natural infections [50]. The current study confirmed this thesis: we found that the antibody level was approximately 1000 times higher after vaccination than after viral infection. A similar observation has been made by many other scientists [42,45,49,51,52,53,54]. Moreover, our data showed slightly higher levels of vaccination antibodies among women and the youngest participants, but there was no association between BNT162b2 vaccination antibodies and sex or age. A similar observation was also made by Moncunill et al., 2021 [49] and Tretyn et al., 2021, [45] while other groups have demonstrated an association between BNT162b2 vaccination antibodies two weeks post the second dose and the sex or age of the participants [54,55,56,57]. Higher antibody titers among women may be explained by stronger type I interferon responses upon stimulation with TLR7 ligands compared to men, which leads to a stronger vaccine response. Weaker vaccine responses in older individuals may be explained by comorbidities; weak immune systems, indicated by high neutrophil to lymphocyte ratios (NLR) and lower type I IFN [42]. It is worth noting that a high NLR is also a poor prognostic factor for COVID-19. Our study showed a significant association between post-infection and post-vaccination antibodies with a clinical course of COVID-19, i.e., a severe course is associated with the highest antibody titer after both infection and vaccination, while asymptomatic and mild infections resulted in the lowest antibody titers. Our findings are in line with previously published studies [49,54,58,59,60].

Finally, the COVID-19 vaccination campaigns assumed two doses of the BNT162b2 vaccine two weeks apart, which should provide a high titer of antibodies protecting the HCWs from severe COVID-19 and hospitalization. Our study presented that approximately six months after the vaccination, the antibody level dramatically decreased. The antibody depletion six to eight months after vaccination was noted also by other authors [49,52,53]. In our study, we did not find any relevant factor associated with the antibody depletion rate, but others found that antibody depletion was faster among non-vaccinated participants than among vaccinated ones [52]. This may lead to the conclusion, that the antibody titers affect the depletion rate, i.e., individuals with asymptomatic and mild infection of SARS-CoV-2 lost their anti-COVID-19 immunity within a few weeks after the disease and are naïve to novel coronavirus reinfections. This highlights the overall importance of vaccination in the protection of our organisms from severe COVID-19. In addition, due to antibody depletion, the third and fourth vaccine dose, the so-called booster dose, should be taken to efficiently protect ourselves from severe SARS-CoV-2 infections. Moreover, the people from the severe COVID-19 risk groups, i.e., individuals over 60 years old, patients after transplantations, patients during anticancer treatment, and patients with immunodeficiencies, should consider a fourth COVID-19 vaccine dose, since their immune system is weaker and antibody titers are relatively lower than in healthy individuals and decreased within a few weeks [61].

To sum up, the biggest strengths of the presented study are (i) longitudinal analysis of the anti-SARS-CoV-2 antibodies among HCWs at four time points over one year, including the time periods after the second and third pandemic wave, as well as time points after COVID-19 vaccination; (ii) simultaneous analysis of the antibodies generated after natural infection (anti-NCP) and antibodies produced as a consequence of vaccination. Besides the above-mentioned strength, we are aware of some limitations of our study. Namely, we analyzed the antibody titers in a relatively small group (N = 90), overrepresented by women and in the case of GaOW overrepresented by individuals over 50 years old, which might bias the results. Moreover, the sample size estimation was extremely hard since no official data on the HCUs employees’ number are provided, and only specific HCU number is known. In addition, it should be noted that the HCU size differs significantly depending on the region and patient number. Moreover, the Polish healthcare system allows HCWs to work in many HCUs in different cities, which complicates the estimation. It should be also highlighted that during the COVID-19 pandemic, the HCWs were very often working on their wards as well as COVID-19 units simultaneously. Therefore, in our opinion, the above-mentioned limitations only strengthen the necessity for further antibody titers analysis on the local and national levels.

## 5. Conclusions

The healthcare workers, who fight in the first line of the COVID-19 pandemic, are at the highest SARS-CoV-2 infection risk. Simultaneously, the healthcare system completely relies on the HCWs. Therefore, the HCW’s herd immunity needs to be constantly analyzed and in case of immunity waning, some extra counter measurement needs to be taken. The best way to protect HCWs against infectious diseases is vaccination. Our study showed that the post-vaccination acquired immunity is stronger and last longer than the post-infection immune response. In addition, we demonstrated that the antibody response after natural infection depends on the clinical course of COVID-19, with the smallest antibody titer in asymptomatic and mild SARS-CoV-2 infections. Thus, it should be highlighted that only COVID-19 vaccination can stop the pandemic, SARS-CoV-2 spread around the globe, new cases and deaths associated with novel coronavirus, as well as the emergence of new genetic variants or virus strains potentially resistant to current vaccines. In addition, our data showed that the HCWs from low infection risk HCUs, such as gynecology and obstetrics wards are at higher infection risk since they lack proper training and expertise on how to handle ‘infectious’ patients. This shows that to combat COVID-19, as well as future pandemics, each HCW needs to be properly prepared, trained and equipped.

## Figures and Tables

**Figure 1 vaccines-10-01576-f001:**
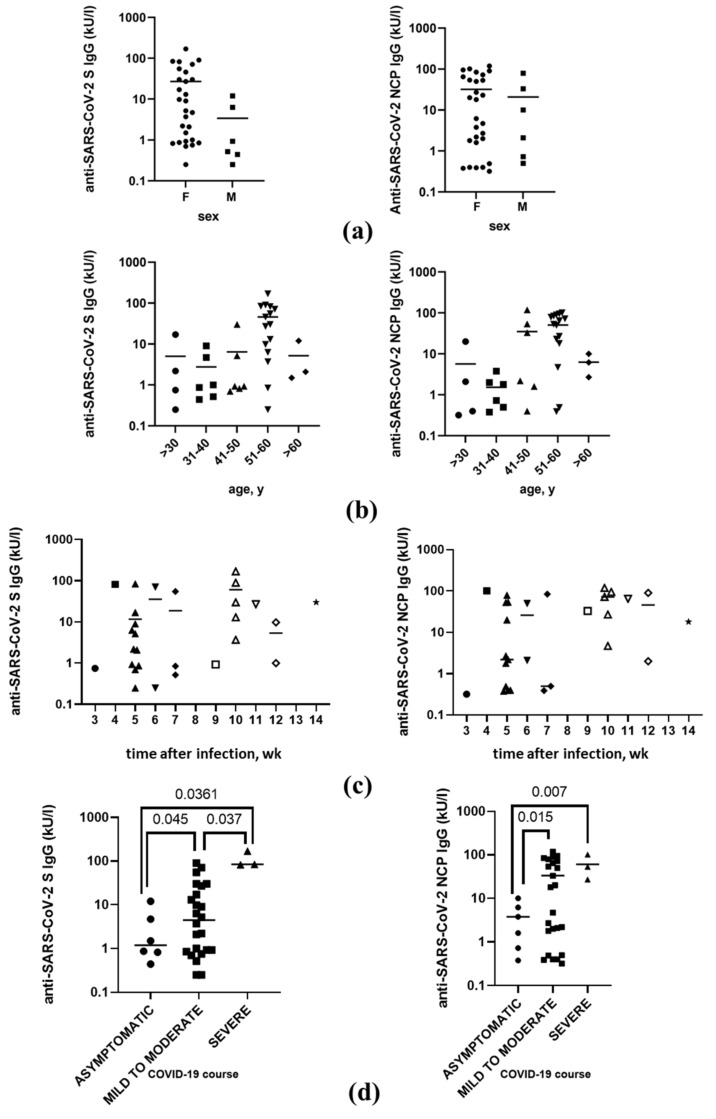
Levels of anti-SARS-CoV-2 spike protein (S, left panel) and anti-SARS-CoV-2 nucleoprotein (NCP, right panel) IgG antibodies among healthcare workers by (**a**) sex, (**b**) age, (**c**) time after SARS-CoV-2 infection, and (**d**) course of SARS-CoV-2 infection. F—female, M—male, y—years, wk—weeks, numbers above the brackets indicate *p*-value; each symbol represents a single study participant.

**Figure 2 vaccines-10-01576-f002:**
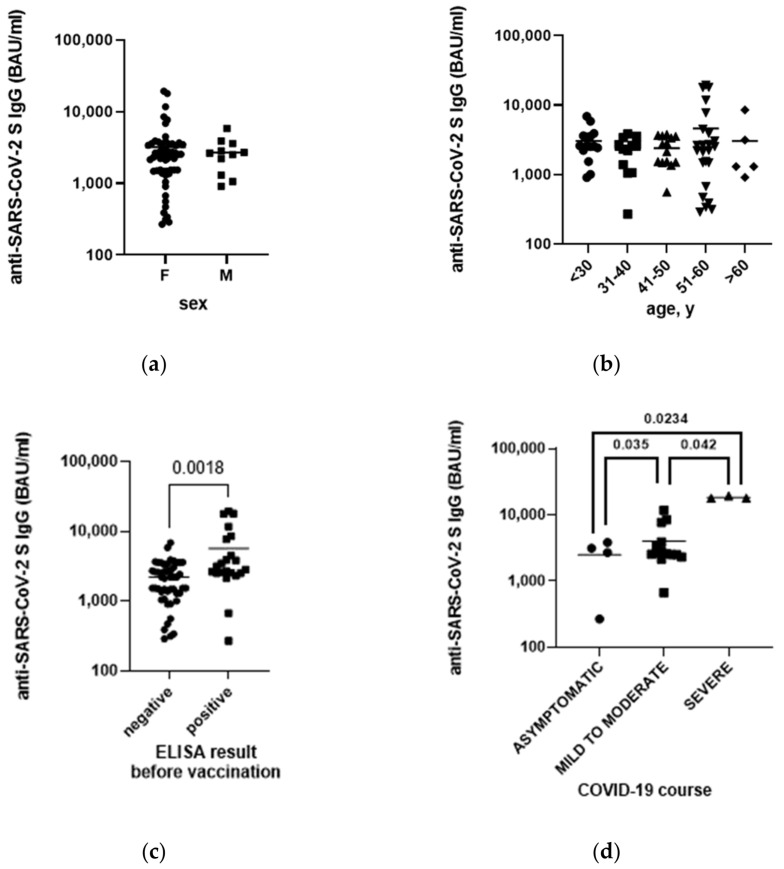
Levels of anti-SARS-CoV-2 spike protein (S) IgG antibodies approximately two weeks after the second dose of BNT162b2 mRNA vaccine by: (**a**) sex, (**b**) age, (**c**) ELISA result before vaccination; (**d**) course of SARS-CoV-2 infection. F—female; M—male; y—years; numbers above the brackets indicate *p*-value; each symbol represents a single study participant.

**Figure 3 vaccines-10-01576-f003:**
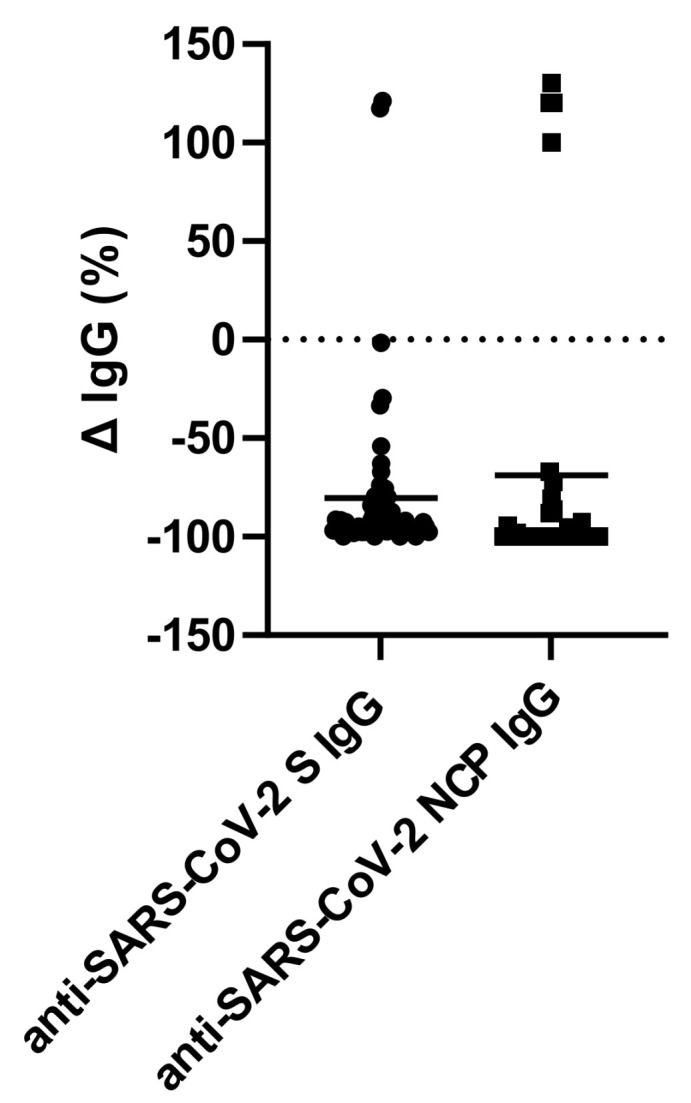
Changes in the anti-SARS-CoV-2 spike protein (S) and anti-SARS-CoV-2 nucleoprotein (NCP) IgG antibody levels six months after vaccination; each symbol represents a single study participant.

**Table 1 vaccines-10-01576-t001:** Characteristic of the analyzed health facility units.

	DIDaCN ^1^	GaOW ^2^
	HCWs	
Study participants	50	40
Female	40	35
Male	10	5
<30 y.o.	16	3
31–40 y.o.	13	2
41–50 y.o.	11	9
51–60 y.o.	9	20
>60 y.o.	1	6
Physicians	16	6
Nurses/Midwives	25	29
Others	9	5
	HCUs	
Hospital beds	20	28 + 3 ^3^
Admitted patients/month in 2020, mean	273	150
COVID-19 patients in 2020	292 + 2443 ^4^	3
SARS-CoV-2 infection risk	HIGH	LOW

^1^ DIDaCN—Department of Infectious Diseases and Child Neurology; ^2^ GaOW—Gynecology and Obstetrics Ward; ^3^ beds + delivery rooms; ^4^ confirmed + suspected cases.

**Table 2 vaccines-10-01576-t002:** The SARS-CoV-2 seroprevalence among the healthcare workers at different time points.

		Seroprevalence (95% CI)			
		September 2020	December 2020	February 2021	September 2021
	N	90	90	90	75
Total	anti-S	0% ^a^ (0.00–4.09)	37.8% ^b^ (28.46–48.10)	100% ^c^ (95.91–100)	89.3% ^d^ (80.34–94.50)
	anti-NCP	nd	nd	26.0% (18.41–35.37)	17.1% (10.28–27.10)
	N	50	50	50	41
DIDaCN ^1^	anti-S	0% ^a^ (0.00–7.14)	26.0% ^b^ (15.87–36.55)	100% ^c^ (92.90–100)	85.40% ^c^ (71.56–93.12)
	anti-NCP	nd	nd	14.0% (6.95–26.19)	5.71% (1.02–18.61)
	N	40	40	40	34
GaOW ^2^	anti-S	0% ^a^ (0.00–8.76)	52.5% ^b^ (37.50–67.07)	100% ^c^ (91.24–100)	94.12% ^c^ (80.91–98.96)
	anti-NCP	nd	nd	38.00% (25.86–51.85)	26.83% (15.70–41.93)

^1^ DIDaCN—Department of Infectious Diseases and Child Neurology; ^2^ GaOW—Gynecology and Obstetrics Ward; N—number of participants; nd—not determined; ^a, b, c, d^—values within a row with different superscript letters differ *p* < 0.05.

**Table 3 vaccines-10-01576-t003:** The SARS-CoV-2 infection risk factors during the second wave of the COVID-19 pandemic.

	Seroprevalence (95% CI)		
	Total	DIDaCN ^1^	GaOW ^2^
Total	37.8% ^ab^ (28.46–48.10)	26% ^a^ (15.87–36.55)	52.5% ^b^ (37.50–67.07)
Sex			
Female	37.3% (27.26-48.65)	22.5% (12.32–37.50)	54.3% (38.19–69.53)
Male	40.0% (19.82–64.25)	40% (16.82–68.73)	40% (7.11–76.93)
Age			
<30	21.0% (8.51–43.33)	25% (10.18–49.49)	0% (0.00–56.15)
31–40	40.0% (19.82–64.25)	30.8% (12.68–57.63)	100% (17.77–100)
41–50	30.0% (14.55–51.90)	27.2% (9.75–56.57)	33.3% (12.06–64.58)
51–60	51.7% (34.43–68.61)	22.2% (3.95–54.74)	65% (43.29–81.88)
>60	42.9% (15.82–74.95)	0% (0.00–94.87)	50% (18.76–81.24)
Occupation			
Physicians	36.4% (19.73–57.05)	31.3% (14.17-55.60)	50% (18.76-81.24)
Nurses/midwives	44.4% ^ab^ (32.00–57.62)	24% ^a^ (11.50–43.43)	62.1% ^b^ (44.00–77.31)
Others	14.3% (2.54–39.94)	22.2% (3.95–54.74)	0% (0.00–43.45)

^1^ DIDaCN—Department of Infectious Diseases and Child Neurology; ^2^ GaOW—Gynecology and Obstetrics Ward; ^a, b^—values within a row with different superscript letters differ *p* < 0.05.

## Data Availability

The data presented in this study are available on request from the corresponding author. The data are not publicly available due to privacy restrictions.

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
