# Peer review of "The Longitudinal Analysis on the Anti-SARS-CoV-2 Antibodies among Healthcare Workers in Poland—Before and after BNT126b2 mRNA COVID-19 Vaccination"

_vaccines, 2022, doi:10.3390/vaccines10101576_

Round 1

Reviewer 1 Report

This manuscript provides evidence concerning rates of SARS-CoV-2 infection among health care workers in two hospital units in Poland and investigates the decay of antibody to infection over time. As such, the evidence is compelling, well-organized, clearly laid out and properly controlled.  The paper does not, however, reach any novel conclusions save in one area that the authors fail to capitalize on. Their novel finding is the much higher rate of infection among nurses and midwives in maternity wards than among physicians or nurses in infectious disease settings. This observation is explained away (lines 282-285), “by the specificity of the work at the GaOW, including not enough time to perform COVID-19 diagnostics when pregnant women have started childbirth, the lower availability of and experience using personal protective equipment, and the uncontrolled behaviour of women during labour contractions.” Much more could be said here, in my view.  In the first place, two other studies have also observed unusually high rates of SARS-CoV-2 infections among maternity ward personnel, including breakthrough infections following vaccination (Alishaq, et al., 2021; Bampoe, et al., 2020). This is a pattern that needs to be addressed. A factor that the authors do not consider is that nurses and midwives are used to handling non-infectious women and may not have the proper training or practice, or even the correct personal protective equipment, to deal with pandemic infections such as SARS-CoV-2 (see, for example, Mbono, et al., 2021; Coxon, et al., 2020; Schmitt, et al., 2021, all of whom document the  lack of knowledge and extensive training or retraining that was required for maternity staff to deal with COVID-19).  Such training, rather obviously, was not needed for personnel working in an infectious disease setting. But much more importantly, the data that the authors provide could be made into a strong call for better infectious disease training (and regular updating) for maternity ward workers, better PPE availability, etc., especially in the context of the ongoing threat of newly emerging pandemic infections (monkeypox!).  There is every reason to believe that maternity ward personnel will forever be at the forefront of dealing with new infectious diseases and this study clearly could be used as a call to make sure that they do not develop two-to-three times the number of such infections in future pandemics.

In sum, I believe that this paper as currently presented is rather humdrum and simply validates many other similar studies; however, I also believe that, with a fairly minor refocusing, the maternity ward data could be made into an extremely important centerpiece that could help to reshape how such personnel are prepared to deal with future pandemics.

Alishaq M, Nafady-Hego H, Jeremijenko A, Al Ajmi JA, Elgendy M, Vinoy S, Fareh SB, Veronica Plaatjies J, Nooh M, Alanzi N, Kaleeckal AH, Latif AN, Coyle P, Elgendy H, Abou-Samra AB, Butt AA. Risk factors for breakthrough SARS-CoV-2 infection in vaccinated healthcare workers. PLoS One. 2021 Oct 15;16(10):e0258820. doi: 10.1371/journal.pone.0258820. PMID: 34653228; PMCID: PMC8519462.

Bampoe S, Lucas DN, Neall G, Sceales P, Aggarwal R, Caulfield K, Siassakos D, Odor PM. A cross-sectional study of immune seroconversion to SARS-CoV-2 in frontline maternity health professionals. Anaesthesia. 2020 Dec;75(12):1614-1619. doi: 10.1111/anae.15229. Epub 2020 Aug 10.

Mbongo JA, Mpika GB, N’dinga H, et al. Midwives’ knowledge of COVID 19 and pregnancy in four maternity units in Brazzaville in 2021. Int J Fam Commun Med. 2021;5(6):187-191. DOI: 10.15406/ijfcm.2021.05.00243

Coxon K, Turienzo CF, Kweekel L, Goodarzi B, Brigante L, Simon A, Lanau MM. The impact of the coronavirus (COVID-19) pandemic on maternity care in Europe. Midwifery. 2020 Sep;88:102779. doi: 10.1016/j.midw.2020.102779. Epub 2020 Jun 10. PMID: 32600862; PMCID: PMC7286236.

Schmitt, N., Mattern, E., Cignacco, E. et al. Effects of the Covid-19 pandemic on maternity staff in 2020 – a scoping review. BMC Health Serv Res 21, 1364 (2021). https://doi.org/10.1186/s12913-021-07377-1

Author Response

Q1 - This manuscript provides evidence concerning rates of SARS-CoV-2 infection among health care workers in two hospital units in Poland and investigates the decay of antibody to infection over time. As such, the evidence is compelling, well-organized, clearly laid out and properly controlled.  The paper does not, however, reach any novel conclusions save in one area that the authors fail to capitalize on. Their novel finding is the much higher rate of infection among nurses and midwives in maternity wards than among physicians or nurses in infectious disease settings. This observation is explained away (lines 282-285), “by the specificity of the work at the GaOW, including not enough time to perform COVID-19 diagnostics when pregnant women have started childbirth, the lower availability of and experience using personal protective equipment, and the uncontrolled behaviour of women during labour contractions.” Much more could be said here, in my view.  In the first place, two other studies have also observed unusually high rates of SARS-CoV-2 infections among maternity ward personnel, including breakthrough infections following vaccination (Alishaq, et al., 2021; Bampoe, et al., 2020). This is a pattern that needs to be addressed. A factor that the authors do not consider is that nurses and midwives are used to handling non-infectious women and may not have the proper training or practice, or even the correct personal protective equipment, to deal with pandemic infections such as SARS-CoV-2 (see, for example, Mbono, et al., 2021; Coxon, et al., 2020; Schmitt, et al., 2021, all of whom document the  lack of knowledge and extensive training or retraining that was required for maternity staff to deal with COVID-19).  Such training, rather obviously, was not needed for personnel working in an infectious disease setting. But much more importantly, the data that the authors provide could be made into a strong call for better infectious disease training (and regular updating) for maternity ward workers, better PPE availability, etc., especially in the context of the ongoing threat of newly emerging pandemic infections (monkeypox!).  There is every reason to believe that maternity ward personnel will forever be at the forefront of dealing with new infectious diseases and this study clearly could be used as a call to make sure that they do not develop two-to-three times the number of such infections in future pandemics.

A1 – We are very grateful for your valuable comments! We modified the Abstract as well as Discussion and Conclusions, according to your suggestions. In the current version of manuscript, we put much more emphasis on the highest seropositivity among the midwives and nurses from the maternity wards, and thus, the necessity of better infectious diseases training, regular updates, and availability of personal protection equipment for the low infection risk healthcare units, which have only occasional contact with infectious patients. We completely agree with Reviewer, that it is extremely important especially in the context of emergence of new pathogens and potential future pandemic.

Q2 - In sum, I believe that this paper as currently presented is rather humdrum and simply validates many other similar studies; however, I also believe that, with a fairly minor refocusing, the maternity ward data could be made into an extremely important centerpiece that could help to reshape how such personnel are prepared to deal with future pandemics.

A2 – We hope that the current version of our manuscript is more interesting and pinpoints the future directions for the healthcare system to be well prepared for the next COVID-19 pandemic waves caused by new SARS-CoV-2 genetic variant/strain, as well as potential future pandemic caused by so far unknown pathogen.

Reviewer 2 Report

I was invited to revise the paper entitled "The longitudinal analysis on the anti-SARS-CoV-2 antibodies among healthcare workers in Poland – before and after COVID-19 vaccination". It was a longitudinal study aimed to evaluate the change in SarsCov2 antibodies titer among HCWs from two different ward in Poland.

Despite the topic is very interesting, the paper presents several criticism:

- Introduction section was too poor. Authors did not described the vaccination plan performed in Poland and when the vaccination campaign started. No mention of vaccine type was also reported. In addition, no similar studies were cited in the background section;

- Sample size estimation was totally missing;

- It is totally unknown the type of vaccine used. Different vaccines can impact differently on IGG titer;

- Authors should describe the methodology used to test IGG titer;

- Authors did not describet the enrollment period. When does it ended?

- It is not clear how many patients were lost to followup. Authors should report the exact number of patients enrolled at each time point;

- Statistical analysis is too poor. Authors should analyze the change in titer by factors (example occurred infection, vaccination, ward ecc). GLM? anova for repeated measures? mixed models?

- Figures should report also p-value of differences among groups;

- In discussions, Authors should add the strenght and limitation section.

After that, in my opinion the main criticism was: how did Authors handled the differences between infected HCWs, vaccinate HCWs and HCWs with both infection and vaccination? It seems that Authors did not take into account this situation the heavly impact IGG titers.

Author Response

I was invited to revise the paper entitled "The longitudinal analysis on the anti-SARS-CoV-2 antibodies among healthcare workers in Poland – before and after COVID-19 vaccination". It was a longitudinal study aimed to evaluate the change in SarsCov2 antibodies titer among HCWs from two different ward in Poland. Despite the topic is very interesting, the paper presents several criticism.

We are very grateful for your valuable comments and suggestions! Simultaneously we hope that we fully answered to your questions and the current version of manuscript will be acceptable for the publication.

Q1 - Introduction section was too poor. Authors did not described the vaccination plan performed in Poland and when the vaccination campaign started. No mention of vaccine type was also reported. In addition, no similar studies were cited in the background section;

A1 – Thank you for your comment! We modified the Introduction section and added required information according to your suggestions Page 2 Lines 55-84)

Q2 - Sample size estimation was totally missing;

A2 – Within our study we examined all employees from the two HCUs and therefore we did not include the sample size estimation. Additionally, we estimated the sample size and analysed HCU representativeness for the Greater Poland region and Poland and included this information as a study limitation within the Discussion section (Page 12 Lines 424-425).

Q3 - It is totally unknown the type of vaccine used. Different vaccines can impact differently on IGG titer;

A3 – Thank you for your valuable comment and please forgive us such an oversight! The information on the used vaccine was included in the sections 2.2. Study design (Page 3 Lines 113) and 3.2. Prevalence of SARS-CoV-2 infection among HCWs (Page 5 Line 187), but we admit that this can be overlooked. Thus, we included the vaccine name in the title of the manuscript (Page 1 Line 3), as well as an Abstract (Page 1 Lines 24) and its short description in the Introduction (Page 2 Lines 74-82).

Q4 - Authors should describe the methodology used to test IGG titer;

A4 – Thank you for your suggestion, however in our opinion the methods, which were used by us to analyse the anti-SARS-CoV-2 level, i.e., ELISA and immunoblots, are routinely used by most of the scientific and diagnostic laboratories and there is no need to describe them in details. In addition, we used the diagnostic kits with the IVD certificates from well-established and reputable manufacturers (EuroImmun GmbH, Germany and Biocheck GmbH, Germany) without any protocol modifications. However, for better understanding we modified the 2.3. Laboratory analysis section (Page 3 Lines 119-131) to explain which antibodies were checked at which time point.

Q5 - Authors did not describet the enrollment period. When does it ended?

A5 – Thank you for your comment! We have contacted the HCUs from the Greater Poland region at the beginning of August 2020 and wait approximately one month for their responses. As it is written in the manuscript, we have got many individuals response from the HCWs from different HCUs with known and unknown SARS-CoV-2 infection risk, as well as specificity of the units. Therefore, we decided to focus on the DIDaCN and GaOW HCWs, since from those two HCUs we got the highest number of responses, the Heads of those units described in details the specificity of the HCU (shown in the Table 1) and we were able to specify the SARS-CoV-2 infection risk. Additionally, we have finished the enrolment of the study participants at the end of August 2020 due to the potential next pandemic wave during autumn 2020, which according to presented data dramatically change the SARS-CoV-2 infection seroprevalence.

Q6 - It is not clear how many patients were lost to followup. Authors should report the exact number of patients enrolled at each time point;

A6 – The required information was included in the 2.2. Study design section (Page 3 Lines 119-131) and Table 2 (Pages 5-6). In three consecutive time points, i.e. September 2020, December 2020 and February 2021, we have tested all enrolled study participants (N=90). At the last time point, i.e. September 2021, 15 out of 90 individuals did not show up without any reasons and did not answer to our contact attempts.

Q7 - Statistical analysis is too poor. Authors should analyze the change in titer by factors (example occurred infection, vaccination, ward ecc). GLM? anova for repeated measures? mixed models?

A7 – Thank you for your suggestion! The additional two-way ANOVA with analysis of the differences among groups by the Tukey post hock test was performed. The data were accepted as statistically different if p<0.05. Additionally, according to the Reviewer suggestion, we analysed the effect of different factors on the antibody depletion rate and did not find any significant difference. The data were included as a Supplementary Figure S2, and description was included in the section 3.5.  Anti-SARS-CoV-2 antibodies level after vaccination (Page 7 Line 270-272)

Q8 - Figures should report also p-value of differences among groups;

A8 – Thank you for your valuable comment. For clarity of the Figures we include only statistically significant differences with p<0.05. Please see Figure 1 and Figure 2.

Q9 - In discussions, Authors should add the strenght and limitation section.

A9 – Thank you for your suggestion! The section on the strength and limitations was included in the Discussion (Page 12 Lines 416-427)

Q10 - After that, in my opinion the main criticism was: how did Authors handled the differences between infected HCWs, vaccinate HCWs and HCWs with both infection and vaccination? It seems that Authors did not take into account this situation the heavly impact IGG titers.

A10 – We would like to thank Reviewer for this comment, but cannot agree that we did not take into consideration the impact of previous SARS-CoV-2 infection on the antibody titers. As shown on Figure 2, and as it was written in the manuscript, one of our key findings is the highest antibody titers among individuals previously infected with novel coronavirus. In addition, we found that the vaccination antibody titer is related with the COVID-19 course, i.e. the lowest antibody level among asymptomatic individuals and the highest level among study participants with severe COVID-19 course. These findings are based on the analysis of two types of anti-SARS-CoV-2 antibodies, i.e. antibodies generated after natural infections (anti-NCP) and vaccination antibodies (anti-S). We carefully analysed the vaccination antibodies titer based on the presence of anti-NCP antibodies and thus we showed the significant differences among above-mentioned groups.

Round 2

Reviewer 1 Report

The revisions much improve the interest and importance of the paper!

Author Response

Q1: The revisions much improve the interest and importance of the paper!

A1:  Thank you very much for your kind comment!

Reviewer 2 Report

Authors improved significantly the manuscript but some point in my opinion remain uncomplete.

- Statistical analysis section contain some mystake: Line 136 - Kruskall Wallis test compares more than two groups; the explanation of two-way anova is poor and it does not explain properly the significance and the aim of this analysis;

- The lack of sample size estimation should be reported as a study limitation and not simply state that "it should be noted that the number of study participants were not high enough to be representative for the whole Poland". I suggest to perform at least a post-hoc power analysis;

- About point 10, Authors did not responded properly: hystory of infection prior the vaccination, between two doses of vaccination or after the vaccination wil affect differently the IGG titers. So it is unclear if Authors take in consideration this point and how it was handled in the multivariate model;

- Figures should report the exact p-values.

Author Response

Authors improved significantly the manuscript but some point in my opinion remain uncomplete.

Thank you for your kind comments! We hope that the current version is suitable for publication in your opinion.

Q1: Statistical analysis section contain some mystake: Line 136 - Kruskall Wallis test compares more than two groups; the explanation of two-way anova is poor and it does not explain properly the significance and the aim of this analysis;

A1: We apologize for the poor statistic description. We deleted the Kruskall-Wallis test from the text. And the two-way ANOVA analysis was explained in details as suggested by the Reviewer (Page 3 Lines 137-144).

Q2: The lack of sample size estimation should be reported as a study limitation and not simply state that "it should be noted that the number of study participants were not high enough to be representative for the whole Poland". I suggest to perform at least a post-hoc power analysis;

A2: We are grateful for your comment. We highlighted the sample size estimation as the biggest study limitation. In addition, we need to admit that the sample size estimation is extremely hard since no official data on the HCU employee number is provided and only the number of specific HCUs are known. Especially the gynecology and obstetrics wards are hard to estimate, since these units exist in each region and differ in size due to the number of admitted patient. In addition, the HCWs are allowed to work in many different units, what even more complicates the estimation. It should be also stated that during the COVID-19 pandemic, HCWs with different specializations worked in the COVID-19 wards. We included these information into the manuscript (Pages 13-14 Lines 438-449).

Moreover, many authors, i.e., Althouse A. Post hoc power: not empowering. Just Misleading. J Surg Res, 2021, 259:A3-6; Goodman SN, Berlin JA. The use of predicted confidence intervals when planning experiments and the misuse of power when interpreting results. Ann Intern Med, 1994, 121:200-206; and Hoenig JM, Heisey DM. The abuse of power: the pervasive fallacy of power calculations for data analysis. Am Stat, 2001, 55:19-24, documented that post hoc power calculations are not useful since the post hoc power calculations are determined by the p-value. According to the above-mentioned experts, this fact leads to the situation when the researcher have the impression that their hypothesized effect may actually exist and they just need a bigger sample size.

Q3: About point 10, Authors did not responded properly: hystory of infection prior the vaccination, between two doses of vaccination or after the vaccination wil affect differently the IGG titers. So it is unclear if Authors take in consideration this point and how it was handled in the multivariate model

A3: We apologize for misunderstanding! Within our manuscript we demonstrated that the patients who underwent SARS-CoV-2 infection and got ELISA positive results, have significantly higher antibody titers in comparison to naïve and ELISA negative study participants (Figure 2C). In addition, we showed that the antibodies titers were related to the COVID-19 course (Figure 2D). But we did not perform additional multivariate analysis, since for example the number of study participants with severe COVID-19 was relatively low, what may bias the results.

Q4: Figures should report the exact p-values.

A4: The exact p-value was added to the manuscript, according to the Reviewer suggestion.

Round 3

Reviewer 2 Report

The paper can now be accepted for publication

Author Response

Thank you!